# The COVID-19 wave in Belgium during the Fall of 2020 and its association with higher education

**Yessika Adelwin Natalia**[1]*, **Christel Faes**[1], **Thomas Neyens**[1,2], **Geert Molenberghs**[1,2]

**1** I-BioStat, Data Science Institute, Hasselt University, Hasselt, Belgium, **2** L-BioStat, Department of Public Health and Primary Care, Faculty of Medicine, KU Leuven, Leuven, Belgium

* yessikaadelwin.natalia@uhasselt.be

**Data Availability Statement:** The data are owned by Sciensano and cannot be made publicly available. Summarized versions are available in the open data repository, https://epistat.wiv-isp.be/

## Abstract

Soon after SARS-CoV-2 emerged in late 2019, Belgium was confronted with a first COVID-19 wave in March-April 2020. SARS-CoV-2 circulation declined in the summer months (late May to early July 2020). Following a successfully trumped late July-August peak, COVID-19 incidence fell slightly, to then enter two successive phases of rapid incline: in the first half of September, and then again in October 2020. The first of these coincided with the peak period of returning summer travelers; the second one coincided with the start of higher education's academic year. The largest observed COVID-19 incidence occurred in the period 16–31 October, particularly in the Walloon Region, the southern, French-speaking part of Belgium. We examine the potential association of the higher education population with spatio-temporal spread of COVID-19, using Bayesian spatial Poisson models for confirmed test cases, accounting for socio-demographic heterogeneity in the population. We find a significant association between the number of COVID-19 cases in the age groups 18–29 years and 30–39 years and the size of the higher education student population at the municipality level. These results can be useful towards COVID-19 mitigation strategies, particularly in areas where virus transmission from higher education students into the broader community could exacerbate morbidity and mortality of COVID-19 among populations with prevalent underlying conditions associated with more severe outcomes following infection.

## Introduction

One year after its start, the ongoing transmission of SARS-CoV-2 still poses a global threat. It was first reported in an outbreak in Wuhan, China [1] and shortly thereafter the resulting corona virus disease (COVID-19) hit the World and was declared a pandemic by the World Health Organization (WHO) [2]. The first wave in Belgium started in March 2020 and decreased importantly towards the summer (late May to early July 2020) [3], and hit particularly hard the elderly population [4]. In August 2020, the number of daily cases increased again, from a 14-day incidence of 11 per 100,000 on July 5, to 71 on August 9. In contrast to

covid/. To access detailed data and replicate our study findings, a data request form found at https://epistat.wiv-isp.be/datarequest/index.aspx has to be returned to the Data Protection Officer office at Sciensano. The authors declare no special access privileges to the data.

**Funding:** C. F. acknowledges funding from the European Union's Horizon 2020 research and innovation programme - project EpiPose (No. 101003688; https://ec.europa.eu/programmes/horizon2020/en). The funders had no role in study design, data collection and analysis, decision to publish, or preparation of the manuscript.

what was seen before, and in line with global trends [5], the incidence increased considerably in the younger age group of 20–29 years old. There are many variants of SARS-CoV-2 circulating in the community and WHO declared some of these variants to be variants of concern (VOC). The Alpha variant started circulating in December 2020, with Beta and Gamma following early in 2021. The Delta variant took off in May 2021, and the Omicron variant in late November 2021 [6].

The increase of COVID-19 incidence also coincide with reopening of educational institution. In Belgium the compulsory education reopening took place on September 1, 2020. However, higher education institutions reopened slightly later on September 14 in the southern Walloon Region and on September 21 in the northern Flanders Region. A large fraction of higher education students in Belgium stays in student accommodation during weekdays, to then return home on weekends. Higher education institutions in the United States had already been identified as locations of increased COVID-19 transmission among young adults [7, 8]. Transmission of COVID-19 was likely facilitated by on- and off-campus congregate living settings and activities [8, 9]. A small-scale survey conducted in a university in Belgium showed that COVID-19 did not influence the commuting behaviour of 47.5% of the students, 40.8% commuted less to home in the weekend, and 8.9% commuted more than before the pandemic [10]. The typical Belgian commuting customs have the potential to exacerbate this phenomenon, and by October 1, 2020, a major increase in incidence occurred.

Another change in COVID-19 transmission could be observed in travelers. While reliable data on incoming travelers are difficult to obtain for many European countries, given the open borders, the German Robert Koch Institute [11], in its epidemiological update on August 18, 2020, reported that one third of German infections trace back to returning travelers. In weeks following, this fraction increased to nearly 50% in Germany, to decline afterwards. Phylogenetic analyses provide further evidence for the impact of international travel [12–14]. These authors have shown that a SARS-CoV-2 variant identified in Spain in early summer 2020, 20E (EU1), subsequently spread across Europe and this spread can only be explained by cross-border traveling.

Socio-demographic factors and population density can also play an important role in COVID-19 transmission. Severely distressed areas with low socioeconomic status in Hong Kong and the United States had higher rates of both COVID-19 cases and fatalities than communities with higher socioeconomic status [15, 16]. Different characteristics were found in New Zealand where almost half of the cases was imported and had higher socioeconomic status [17]. The latter can be explained by the very low incidence in this island nation throughout the pandemic, such that the inclusion of a relatively small number of cases can affect the overall incidence substantially. In general, infectious diseases spread more rapidly in densely populated areas. In very populous countries such as India and Algeria, population density had a direct impact on COVID-19 transmission [18, 19]. A comparative study among countries in Europe, East Asia, Australia, and the United States reported similar results [20]. In this paper, we will, in addition to the impact of the student population, investigate the impact of the socioeconomic status and population density on the COVID-19 incidence in Belgium.

This study is aimed at exploring and contextualising the evolution of COVID-19 incidence in Belgium from June until December 2020. We evaluated the effect of higher education students on the age-specific COVID-19 incidence, as well as differences according to gender, socioeconomic status, and population density. We make use of spatial conditional auto-regressive Poisson models, that account for the spatial association amongst regions.

## Materials and methods

### Data

Individual data of daily COVID-19 confirmed cases, along with their gender, age and residential municipality, were collected and made available by Sciensano, the Belgian institute for public health [21]. We retrieved the data from 1 June until 31 December 2020. The Alpha variant started circulating at the very end of this period, but in low percentages only. All other variants of concern emerged only in 2021. Belgium is subdivided in 581 municipalities. In order to present our results comprehensively, we used Belgian map made available online by Statbel, the Belgian national statistics institute [22]. To explore the effect of other variables, we also used information on the number of students in the higher education, population data, and mean income per municipality as proxy for socioeconomic status provided by Statbel. We obtained population data for the year 2020. However, the last census count of the student population was conducted in 2016, which we used to calculate the number of students and the mean income per municipality.

### Statistical methods

A spatially discrete geostatistical model is fitted to the incidence data originating from separate time intervals and sub-regions. We use 15-day intervals and investigate two sub-regions (Flanders and Wallo-Brux), which allows us to flexibly investigate dynamics at specific points in time within geographical entities, in which transmission dynamics are expected to behave differently.

Within each region, consisting of municipalities $i$ ($i = 1, \ldots, I$), and per time period of two weeks, $O_{ikl}$ denotes the number of observed COVID-19 cases in municipality $i$, for age group $k = 1, \ldots, 8$ corresponding to age groups 17 years or younger, 18–29 years, 30–39 years, 40–49 years, 50–59 years, 60–69 years, 70–79 years, and 80 years or older; for gender $l = male, female$. $N_{ikl}$ represents the population size per municipality, age group, and gender.

We model $O_{ikl}$ using a Poisson regression model

$$O_{ikl} \sim \text{Poisson}(N_{ikl}\theta_{ikl}) \tag{1}$$

with $\theta_{ikl}$ the incidence of COVID-19 cases in municipality $i$, for age group $k$, and gender $l$, which is modelled on the logarithmic scale as

$$\eta_{ikl} = \log(\theta_{ikl}) = \alpha_{1k} + \alpha_{2l} + \beta_1 \texttt{income}_i + \beta_2 \texttt{log(popdens)}_i + \gamma_k * \texttt{sturatio}_i + v_i + \upsilon_i \tag{2}$$

with parameters $\alpha_{1k}$ and $\alpha_{2l}$ representing the means of age and gender groups. Municipality-specific covariates used in the model are: standardized mean income (as a proxy for the socioeconomic status of the municipality) denoted as income$_i$, logarithm of population density per km$^2$ denoted as log(popdens)$_i$, and standardized student ratio, i.e., the number of higher education students per 100 inhabitants in area $i$ denoted as sturatio$_i$. Note that we assume an age-specific parameter $\gamma_k$ for the student ratio, as interest is in the impact of higher education students on the incidence in each age group.

The model allows for both spatially structured heterogeneity and unstructured heterogeneity, via the inclusion of two random effects ($v_i$ and $\upsilon_i$) [23]. The spatially-unstructured random effect $\upsilon_i$ is assumed to follow an independent normal distribution $\upsilon_i \sim N(0, \sigma_\upsilon^2)$. The spatially-structured random effect $v_i$ is defined as a Gaussian random field, accounting for the spatial autocorrelation. Each municipality $i$ is characterized by a set of neighbours $\mathcal{N}_i$, defined as the areas that share boundaries with municipality $i$. The intrinsic conditional autoregressive (CAR) model specifies the distribution of $v_i$, conditional on all the other values $v_j$ for $j \neq i$, is

given by:

$$v_i | v_{-i} \sim N(\bar{v}_i, \sigma_i^2)$$

$$\bar{v}_i = \frac{1}{|\mathcal{N}_i|} \sum_{j=1}^{n} a_{ij} v_j,$$

$$\sigma_i^2 = \frac{\sigma_v^2}{|\mathcal{N}_i|},$$

(3)

where the weights $a_{ij}$ are defined as 1 if areas $i$ and $j$ are adjacent and 0 otherwise; and $|\mathcal{N}_i|$ denotes the number of neighbors of of area $i$. This model specification, with a spatially structured and an unstructured random effect, is commonly described as a convolution model.

The CAR convolution models were fitted using integrated nested Laplace approximation (INLA). A wide and flexible class of models ranging from generalized linear mixed models to spatial and spatio-temporal models can be implemented with INLA. It eases computation when performing approximate Bayesian inference in latent Gaussian models compared to traditional Markov chain Monte Carlo (MCMC) methods [24]. The analysis was performed in R 3.6.3 [25], through the package R-INLA [26].

## Ethics statement

All data used in this study were anonymous and therefore did not allow us to identify patients. The data are owned by Sciensano and cannot be made publicly available. Summarized versions are available in the open data repository, https://epistat.wiv-isp.be/covid/. To access detailed data and replicate our study findings, a data request form found at https://epistat.wiv-isp.be/datarequest/index.aspx has to be returned to the Data Protection Officer office at Sciensano.

## Results

Fig 1 depicts how the moving seven-day sum of confirmed cases over the most recent period increases or decreases relative to the immediately preceding seven-day period. It shows a rapid increase in early September, arguably due in part to incoming travelers at the end of summer holidays at the end of August, with compulsory education starting on 1 September 2020.

There were 67,292 cases reported with known residential, age, and gender information from 1 July until 31 December 2020. The cases were categorized into 8 age groups as shown in Fig 2. The number of daily reported cases increased strongly, starting from the beginning of September 2020 with the highest incidence at the end of October 2020.

As shown in Fig 3a, the municipalities in Belgium can be grouped according to the Region: Flanders, Walloon, and Brussels. For the analysis we grouped Brussels, the Capital Region, together with the Walloon Region due to the main language being used there, and termed Wallo-Brux (Walloon is exclusively and Brussels predominantly French speaking).

The distribution of each variable per municipality is shown in Fig 3. The student ratio is presented in Fig 3b. The average student ratio is 3.51 students per inhabitant and ranges from 0.82 to 10.49 students per inhabitant. We could see that the student ratio is higher in areas closer to university towns. On the other hand, there is a north-south trend, with lower student ratio in the south of the country. The yearly mean income per municipality is presented in Fig 3c. The mean income ranges from €8,835 to €28,348, with an overall average of €18,465. In this map, differences amongst Flanders and Wallo-Brux are visible. The population density (Fig 3d) ranges from 0.05 tot 171.79 inhabitants per $km^2$, with an overall average of 7.44 inhabitants per $km^2$. This map clearly shows the more densely populated areas.

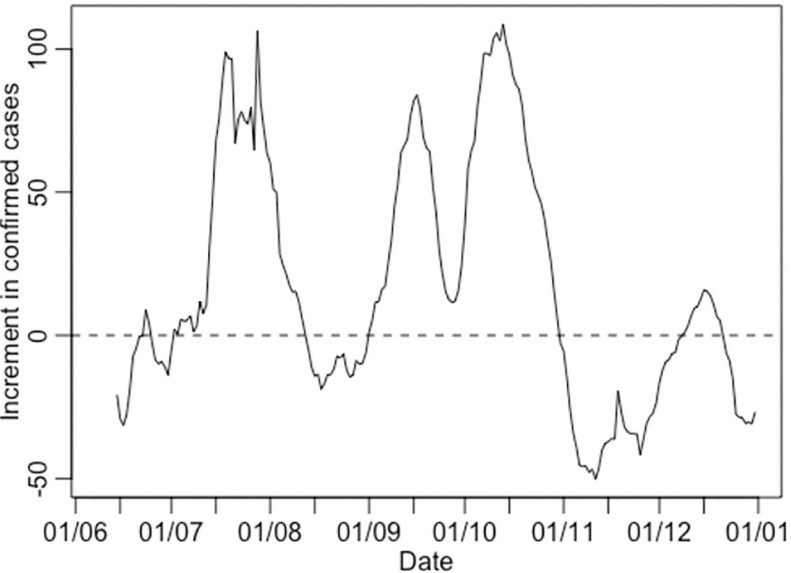

**Fig 1. Increment function of COVID-19 confirmed cases.** The % increase or decrease in the rolling seven-day sum of the number of cases is relative to the immediately preceding seven-day period. The curve starts on June 1, 2020 and runs through the end of 2020.

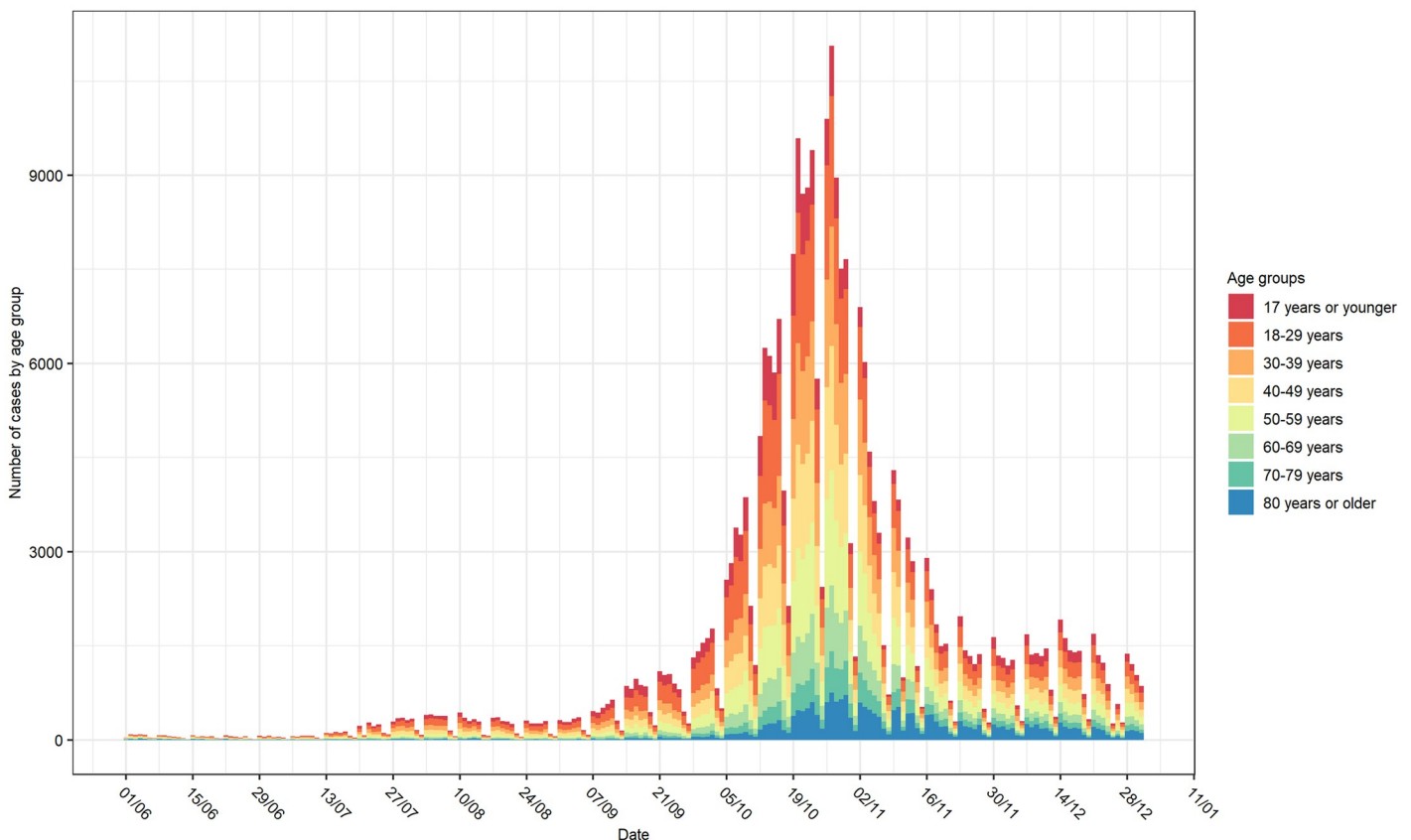

**Fig 2. Distribution of COVID-19 cases per age group from 1 June until 31 December 2020.**

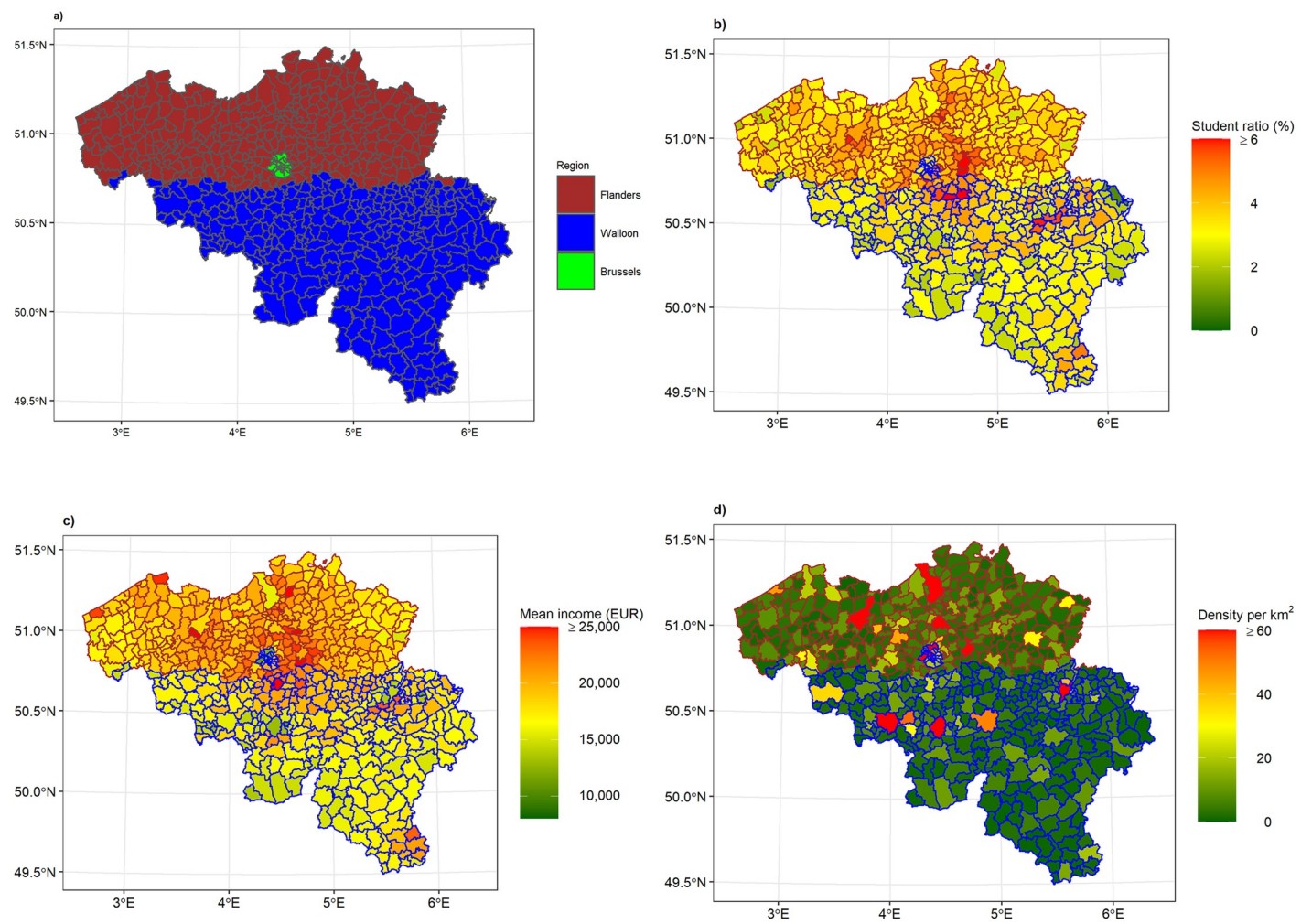

**Fig 3. Descriptive map of Belgium.** The regional division is depicted in (a). Distribution per municipality of student ratio, yearly mean income, and population density are depicted in (b), (c), and (d), respectively. The red border in (b), (c), and (d) indicates the Flanders Region and the blue border indicates the Brussels and Walloon Regions.

The growth in incidence and positivity rate is also geographically different. While the growth rate of the municipality-level incidence in August shows increases in the central part of the country, in October, it shows a clear north-south divide (Fig 4). A similar picture between both sides of the country emerges from the growth in positivity ratio (S1 Fig).

To offer a different perspective on COVID-19 evolution in Belgium, we summarized the number of reported COVID-19 cases with preventive measures taken in Flanders and Wallo-Brux in Fig 5. Considering similar preventive measures taken in these two regions, we observed considerable difference in the number of reported cases before and after September 2020. The number of COVID-19 cases in Wallo-Brux increased earlier and reached a higher level when compared to Flanders, following the reopening of higher education.

## Age differences

The period- and region-specific overall incidence per age group, estimated by $\exp(\alpha_{1k})$, is presented in Fig 6. We observe that the age group of 18–29 years, followed by age group 30–39 years had the highest predicted incidence of COVID-19 cases, particularly from mid-

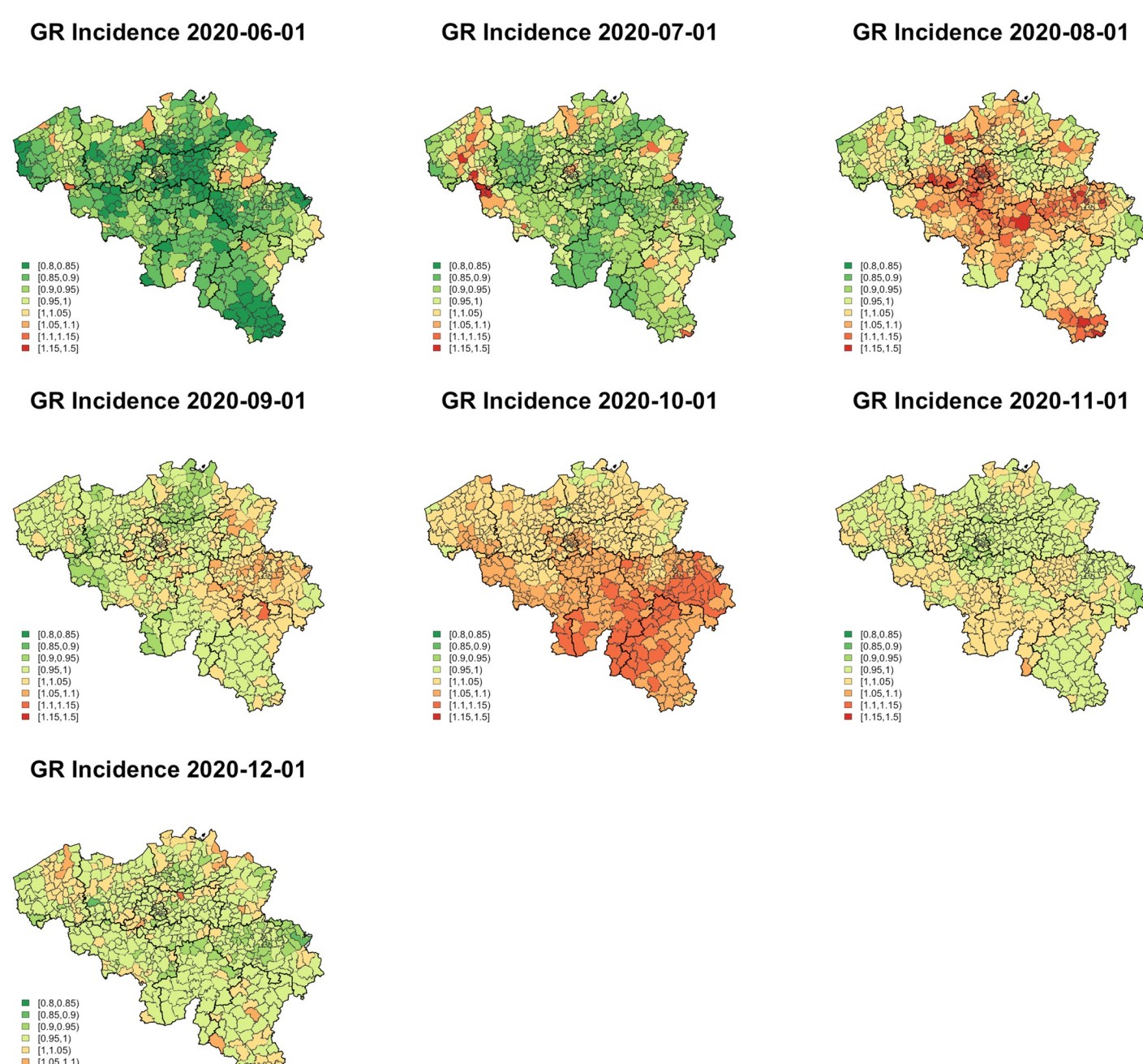

**Fig 4. Municipality-level growth rate in incidence between June and December 2020.**

September 2020 until the end of October 2020, in both Flanders and Wallo-Brux. From November 2020 onwards, the predicted incidence was highest within the group of those older than 80 years in both regions. This suggests that this epidemic wave started in the younger individuals, and that infection spread from the younger individuals to the elderly population. While the trend looks similar in Flanders and Wallo-Brux, note that the incidence in the latter is much higher at the time of the peak and that the effect of age group was less pronounced in Flanders. Indeed, the fact that younger age groups are less likely to present for testing, and also

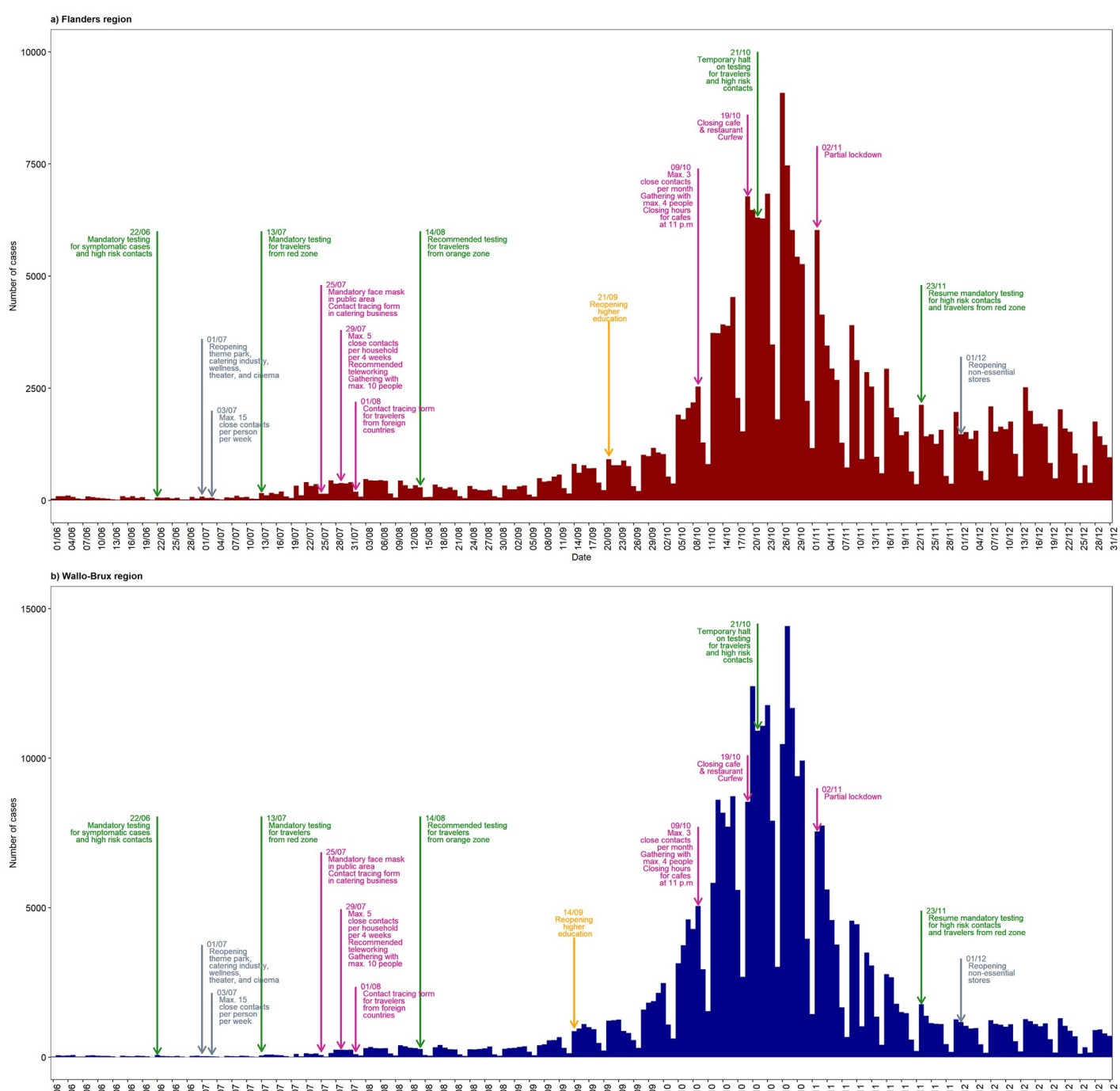

**Fig 5. COVID-19 epidemic curve.** Different colors indicate different type of strategies taken in this period. Green = testing strategy, grey = relaxation strategy, purple = preventive measures.

that they are nevertheless very prominently present among confirmed cases in September-October 2020, underscore this point. We also note that there is no evidence for variants-of-concern circulating over the study period. The alpha variant was identified in Belgium in December 2020 for the first time.

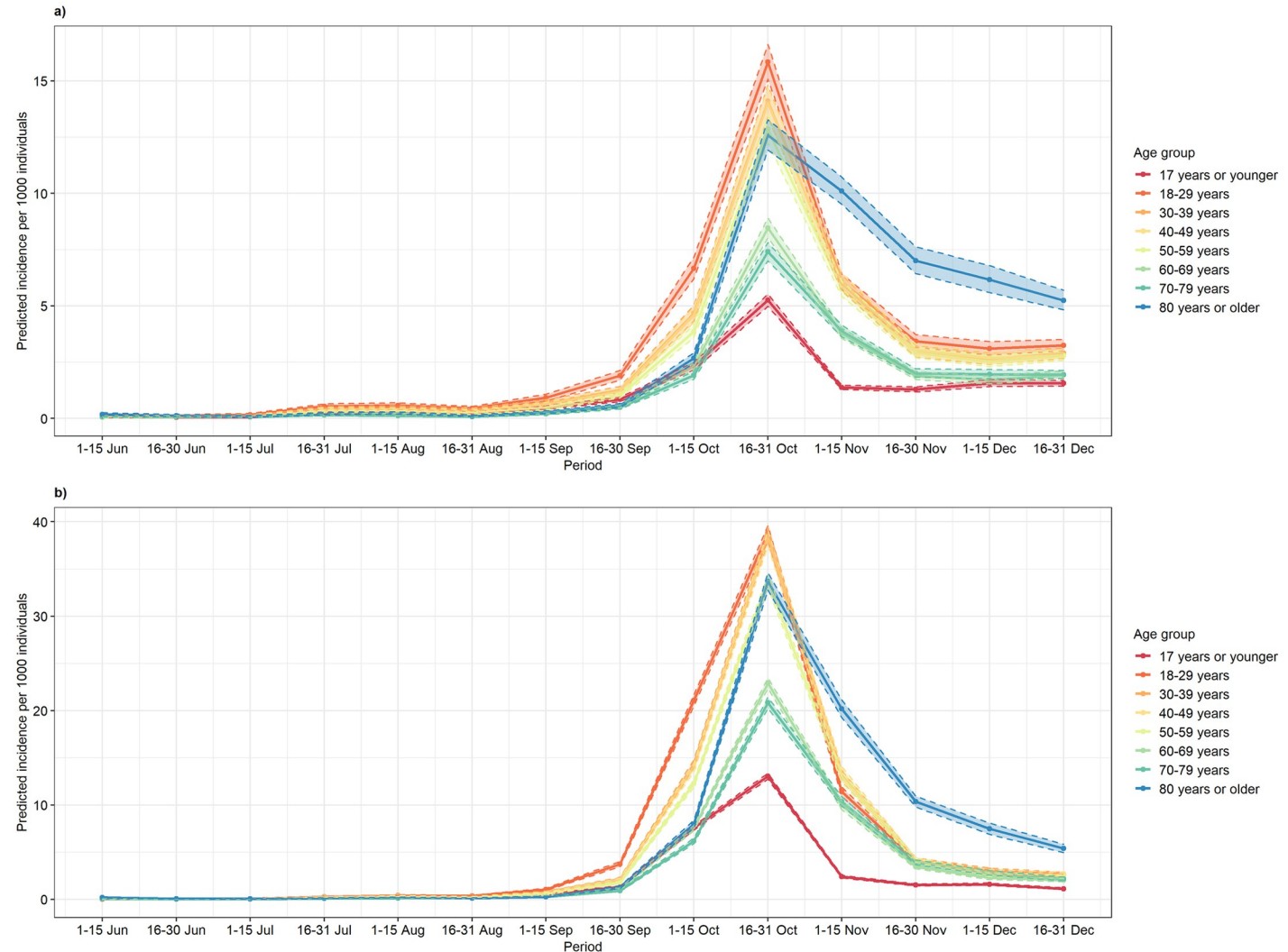

**Fig 6. Effect of age group to predicted COVID-19 incidence.** a) Flanders Region, b) Wallo-Brux Region.

## Socio-demographic effects

Fig 7 shows the multiplicative effect of other variables on the incidence for Flanders (Fig 7a) and Wallo-Brux (Fig 7b). In Flanders, the predicted incidence of COVID-19 cases increased above 1 in females starting from the beginning of September, while this effect was always high in Wallo-Brux. A similar increasing trend can be observed for the population density in Flanders, particularly in areas with low population density. Areas with higher population density in Wallo-Brux had higher incidence in June-September 2020 and then decreased to below 1 from October 2020. While regions with a lower population density had a lower incidence during summer months, the difference between municipalities with low and high population density largely disappeared during the October-November wave. Differences in incidence amongst municipalities with higher or lower socio-economic status (reflected by the mean income of the municipality), shows a fluctuating trend. In Flanders, the effect of mean income was above 1 (meaning that areas with a higher socio-economic status have a higher incidence) in the beginning of summer and then decreased to below 1. On the other hand, the effect of mean

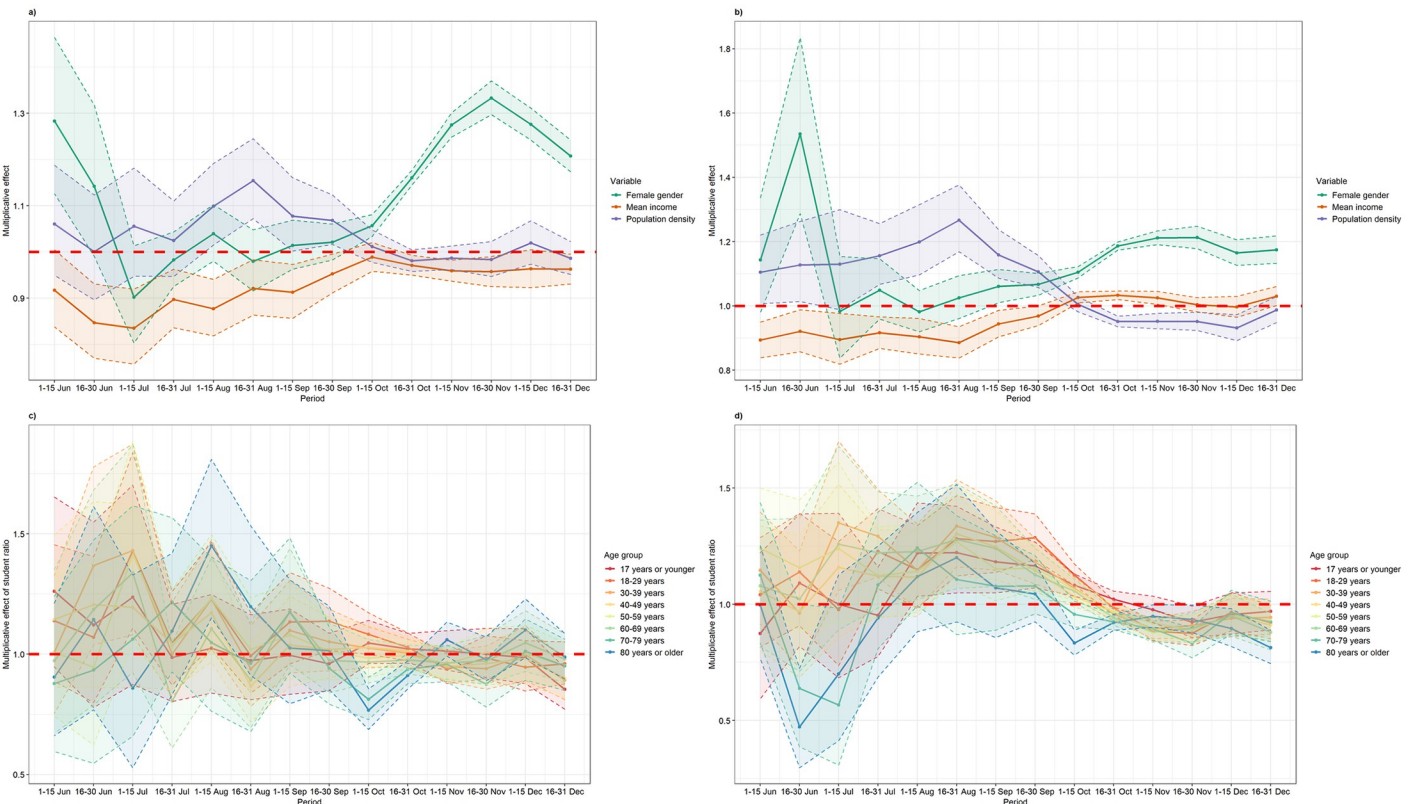

**Fig 7. Effect of other covariates.** Top: Effect of gender, mean income, and population density to predicted COVID-19 incidence in Flanders (a) and Wallo-Brux (b); Bottom: Multiplicative effect of higher education students to COVID-19 incidence in Flanders (c) and Wallo-Brux (d).

income in Wallo-Brux was relatively low in the summer months and then increased to above 1 in October-November 2020.

Looking at the effect of the student population in each municipality as presented in Fig 7c and 7d, we observe that the presence of higher education students increased the local incidence of COVID-19, particularly in the age groups 70–79 years and 80 years or older in Flanders during the summer months and the effect decreased afterwards. Different trends were found in other age groups where the multiplicative effect showed a slight increase in younger age groups throughout the study period. In Wallo-Brux, the multiplicative effect of higher education students showed an increasing trend towards September 2020 in all age groups, particularly in age groups 18–29 years and 30–39 years, with the highest peak in mid-August 2020.

## Geo-spatial effects

The map of predicted COVID-19 incidence per municipality based on confirmed cases between 1 June and 31 December 2020 is shown in (Fig 8). In the summer months, the predicted incidences were very low and increased significantly in October 2020 in all regions. The highest predicted incidence was found in Wallo-Brux during the second half of October 2020, ranging from 361 to 8,633, with an overall average of 3,132 per 100,000 population. The predicted incidence significantly decreased in the second half of November 2020. However, some municipalities in both regions still maintained high predicted incidences until the end of the year.

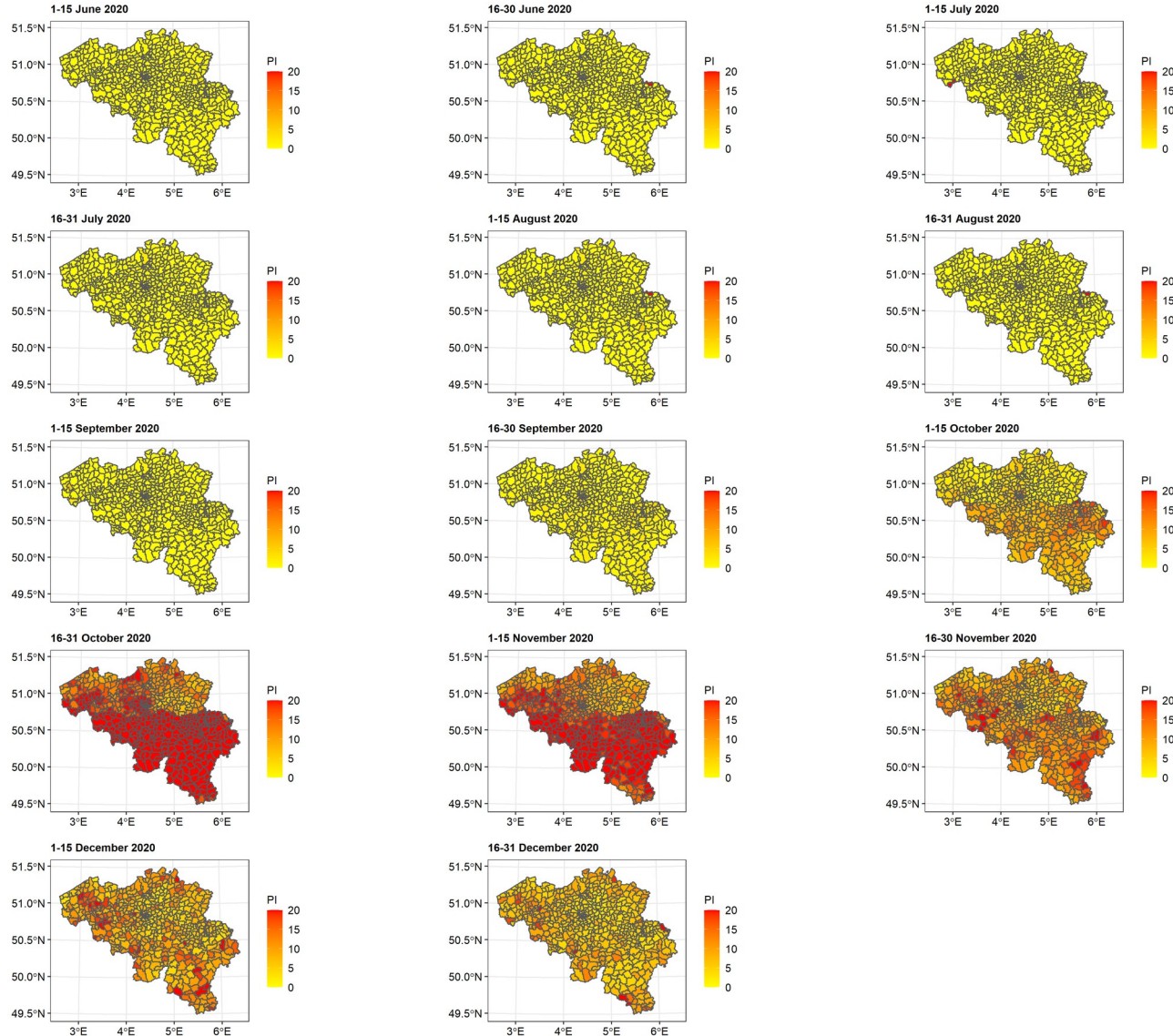

**Fig 8. Predicted COVID-19 incidence per 1000 population at municipality level.** PI = predicted incidence.

Fig 9 shows the distribution of spatial random effects. The results support the hypothesis of a non-uniform distribution of COVID-19 incidence across regions.

## Discussion

During the Spring 2020 COVID-19 wave, Belgium sustained 10,443 deaths, of which about 65% occurred in the nursing home population [27]. In the second wave, which we have investigated, the transmission shifted at first to younger age groups and was associated with less fatalities [28]. Other countries also reported that the COVID-19 pandemic wave during the Fall of 2020 affected a larger proportion of younger adults as compared to the first wave in March 2020 [29–31]. Younger adults are more likely to have mild or no symptoms and they can unknowingly contribute to COVID-19 transmission in the population, which potentially jeopardizes people at higher risk for severe illness [32, 33].

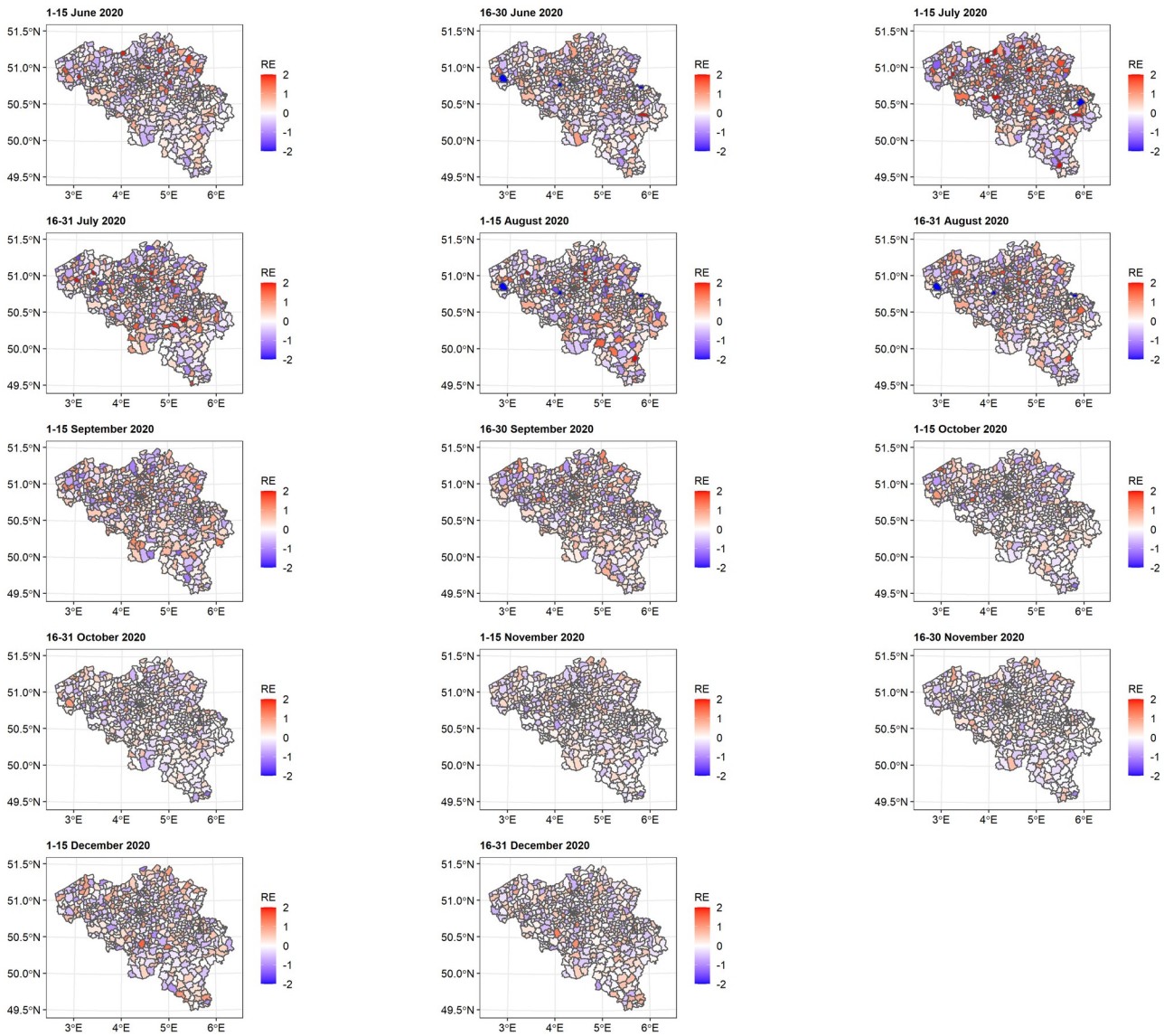

**Fig 9. Distribution of spatial random effect at municipality level.** RE = random effect.

As education resumed with contact learning for the academic year of 2020–2021, it was important to monitor trends in COVID-19 transmission among students. After reopening higher education in September 2020, COVID-19 incidence among adolescents aged 12–17 years was approximately twice that in children aged 5–11 years in the United States [34]. Similar phenomena were also reported in the Reggio Emilia province in northern Italy, where transmission within schools occurred particularly among adolescents aged 10 to 18 years, i.e., those enrolled in secondary and high schools [35].

Compared to students in compulsory education, university students have more elaborate social mixing patterns, which play an important role in COVID-19 transmission. Many university students in Belgium commute, as well during the period under investigation in our study, from home where they still live in the same environment with older people such as parents and grandparents. This increases the possibility of transmitting the infection to

those who are at increased risk for severe disease [36]. Moreover, younger adults are more likely to engage in "high risk" behavior. Some studies, conducted during the initial phase of the COVID-19 pandemic (March–April 2020), show that younger adults were more likely to leave their homes frequently [37] and take part in social activities such as parties, fitness, or casual hang-outs [38–41]. Note that we expect this to be less the case for children. Moreover, it has been documented that parents during the pandemic's first wave often protected their children against important transmission routes, even when they would continue to have work-related and social contacts [42]. Inappropriate practice of preventive measures such as wearing facial masks and social distancing, is more common in this age group as well [43]. This is expected to render the preventive measures less effective and will likely boost further virus transmission. Clearly, higher-education students belong to an age group that partly consists of working individuals [44], which a priori might complicate understanding dynamics in COVID-19 transmission in this age group, particularly after reopening of higher education. However, the most recent relaxations, taking place prior to the start of the academic year, did not address the workplace and teleworking had been recommended uninterruptedly since July 2020; further relaxation in economic activities did not occur until December 2020 (Fig 5). Based on this, we expect that the impact of behavior in the working sub-population within this age group on changes in COVID-19 incidence rates is minimal.

For reasons stated above, we focus on higher education alone, rather than on compulsory education as well. There are two more arguments to justify this. First, incidences in the compulsory school population, monitored directly by the school system itself have been considerably lower than that of the general population, from the start of the school year (September 1, 2020) until February 2021. For example, on November 1, 2020, around the peak of the second wave, the incidence in the Flemish population was 1,118, while that in the compulsory school system, among pupils, 355. Second, compulsory education is also more homogeneously spread throughout the population (both in terms of school location as geographical spread of pupils), precisely because of its compulsory nature.

We found increased predicted incidence of COVID-19 among females. The gender effect could be explained by the demographic structure of Belgium where females comprise slightly more than half of the population [45]. Another, likely more important, element is that women constitute a high percentage of caregivers in both the formal and informal sectors, which increases the risk of contracting COVID-19 [46]. We also found an increased multiplicative effect of highly populated areas in Wallo-Brux. However, a negative association is observed from October 2020 onwards. This suggests that the transmission might have shifted to less populated areas. Of note, some less populated areas in Denmark had an incidence closer to that of the capital area [47].

We observed differential effects of mean income between the regions, i.e., Flanders and Wallo-Brux, and increasing trend of mean income in Wallo-Brux, with the reverse holding in Flanders. An increased effect of mean income combined with increased predicted incidence after the summer months implies that, from September onwards, more COVID-19 cases have been reported in areas with higher socioeconomic status. This finding further suggests that more COVID-19 cases during the Fall wave were detected in returning travelers from summer holidays. A man in Iceland was diagnosed with COVID-19 after vacationing abroad [48]. A genomics study in Australia reported that importations by returned international travelers accounted for over half of locally acquired cases [49]. Minimizing non-essential travel such as holiday is important towards reducing the importation and reduced the outbreak size [14]. Of importance of Europe is the rapid spread of 20E (EU1) strain across the continent in the late summer of 2020 [12].

There are some limitations of our study that must be considered. First, analyzing data at municipality level could minimize differences in age group–specific trends that might be observed at smaller area level. Second, dynamic population activity data would provide a better picture of the impact of human activities on COVID-19 transmission. The presence of this population activity data in the model may help fine-tune preventive measures in the future. Third, in order to flexibly model temporal changes, we now chose to fit separate spatial models for each two-week interval. One important disadvantage of this approach is that it assumes the absence of temporal correlation at the municipality level. However, the implementation of a spatio-temporal model yielded time trends that were overly smoothed. We are investigating methods to overcome this problem in the future. Fourth, there are spatio-temporal dynamics in COVID-19 testing strategies and test capacity [50]. We therefore analyzed the data per region, since these dynamics were mainly driven by COVID-19 preventive measures, a considerable part of which was based on decisions taken by regional authorities. Over a long period of time, encompassing the study period, COVID-19 testing has targeted possible COVID-19 cases (symptoms), high-risk contacts (e.g., as identified via contact tracing), travelers, patients prior to receiving medical care, etc. An important exception is the period from late October 2020 to late November 2020 during which, because of very high circulation (up to 20,000 confirmed cases per day), asymptomatic cases were no longer tested. We found a clear North-South divide in the growth rate (Fig 4) as well as the positivity ratio (S1 Fig). Sciensano reported that the number of COVID-19 tests performed in the Flemish Region was 22.9% higher when compared to Wallonia and Brussels in this period [51]. There is a possibility that many asymptomatic or mild cases could not be detected in Wallo-Brux. Evidently, without proper quarantine procedures, the infected person could further transmit the virus in the population.

Last but not least, this study was conducted prior to the wide-scale deployment of COVID-19 vaccination. Therefore, the results cannot be interpreted as directly representative for a vaccinated population. The Belgian government started the vaccination campaign in late December 2020, with an initial focus on nursing home residents, healthcare workers, residents with co-morbidity, and then the general population in decreasing age bands. In late August 2021, a bit over 70% of the Belgian population is fully vaccinated with adult coverage (18+) of 83% [52]. Vaccination is one of the primary mitigation strategies to avoid future higher-education induced COVID-19 waves. Even though current vaccination coverage is satisfactory, [53] reported that levels of willingness to accept a COVID-19 vaccine in many countries are insufficient to meet the requirements for herd immunity or even for a vaccination level that obviates the more stringent non-pharmaceutical interventions, underscoring the importance of the latter, particularly when transmission of COVID-19 is still relatively high in some populations.

The dynamics of COVID-19 transmission changes rapidly and at the time of writing (December 2021), the Delta followed by the Omicron variants are the main circulating variants globally [6]. In Belgium, more than 90% of the infection is caused by the Delta variant since late July until early December 2021. Consequently, COVID-19 incidence in Belgium is also increased considerably due to the increased transmissibility of this variant. During Fall wave of 2020, which was analysed here, COVID-19 incidence in the compulsory school population was considerably lower than in the general population, while this was not the case for the Fall wave of 2021, when incidence in children aged 0–9 years old was reported to be 5.3 times higher than during the same period in 2020. For adults between 20–49 years old, the COVID-19 incidence was increased around 1.04 times [54]. There are many factors that contribute to the virus transmission. However, we observed similar trend in year 2020 and 2021, i.e. a considerable increase of COVID-19 incidence after the beginning of the new academic year, despite the shift of transmission dynamics in younger children. This underscores the relevance of our results in the new transmission scenario.

## Conclusion

The results in this study provide insights into the association between reopening higher education in the academic year 2020–2021 and increased COVID-19 incidence in the Fall 2020 wave. These findings have important public health implications. Reopening higher education should be carefully planned, particularly when incidence is high and/or in areas with large proportion of high-risk individuals. Cooperation between academic and local authorities is needed to strengthen community mitigation. Increasing COVID-19 testing in students might be considered as a manner to rapidly detect asymptomatic cases in this age group.

## Supporting information

**S1 Fig. Municipality-level growth rate in positivity ratio between June and December 2020.**
(TIF)

**S2 Fig. Standardized incidence rate per 100,000 population at region level.**
(TIF)

**S3 Fig. Predicted COVID-19 incidence per 1000 population at municipality level.** Different legend scales were used to accommodate the difference between two regions before and after September 2020.
(TIF)

**S1 Table. Parameters estimation.** Estimation of all parameters used in the spatially discrete geostatistical model.
(XLSX)

## Author Contributions

**Conceptualization:** Christel Faes, Geert Molenberghs.

**Data curation:** Christel Faes, Thomas Neyens.

**Formal analysis:** Yessika Adelwin Natalia.

**Funding acquisition:** Christel Faes.

**Investigation:** Yessika Adelwin Natalia.

**Methodology:** Christel Faes, Thomas Neyens, Geert Molenberghs.

**Software:** Yessika Adelwin Natalia.

**Supervision:** Thomas Neyens, Geert Molenberghs.

**Validation:** Thomas Neyens, Geert Molenberghs.

**Visualization:** Yessika Adelwin Natalia, Thomas Neyens.

**Writing – original draft:** Yessika Adelwin Natalia.

**Writing – review & editing:** Christel Faes, Thomas Neyens, Geert Molenberghs.

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
