## [Decision Letter · Decision Letter 0]

9 Aug 2021

PONE-D-21-17289

The COVID-19 wave in Belgium during the Fall of 2020 and its association with higher education

PLOS ONE

Dear Dr. Natalia,

Thank you for submitting your manuscript to PLOS ONE. After careful consideration, we feel that it has merit but does not fully meet PLOS ONE’s publication criteria as it currently stands. Therefore, we invite you to submit a revised version of the manuscript that addresses the points raised during the review process, which you can find below, in the end of this email.

We look forward to receiving your revised manuscript.

Kind regards,

Oana Săndulescu

Academic Editor

PLOS ONE

Journal Requirements:

4. We note that Figures 2,4, 7, 8 & S1-S3 in your submission contain [map/satellite] images which may be copyrighted. All PLOS content is published under the Creative Commons Attribution License (CC BY 4.0), which means that the manuscript, images, and Supporting Information files will be freely available online, and any third party is permitted to access, download, copy, distribute, and use these materials in any way, even commercially, with proper attribution. For these reasons, we cannot publish previously copyrighted maps or satellite images created using proprietary data, such as Google software (Google Maps, Street View, and Earth). For more information, see our copyright guidelines: http://journals.plos.org/plosone/s/licenses-and-copyright.

a. You may seek permission from the original copyright holder of Figures 2,4, 7, 8 & S1-S3 to publish the content specifically under the CC BY 4.0 license.  

Additional Editor Comments:

Thank you for submitting your work to PLoS One. This is an important paper, which I feel could be further improved by addressing the comments below from myself and from the expert peer-reviewers:

- The Introduction section is quite long and includes some information that might be better suited to the Discussions section.

- Lines 21-23: In the beginning of the pandemic, sequencing was not routinely performed in all countries. Therefore, what was considered to be “a Spanish strain” might not have been Spanish in origin, but only undetected in countries that had not implemented sequencing yet. I would recommend revising this categorical statement that its spread “can only be explained by cross-border travelling.”

- Same comment for lines 228-229.

- Lines 33-35: Is there data to confirm or deny whether “The typical Belgian commuting customs” were maintained throughout the studied period in this paper? Many students in many countries changed their commuting customs either due to travel restrictions, or due to personal reasons, for example to avoid visiting elderly relatives or relatives with comorbidities.

- Lines 62-63: There is no need to include this statement, since all articles are structured this way: “This is followed by the results and conclusion from this study.”

- Some of the information presented in the Methods section might be a better fit in the Results section.

- Please define the variable abbreviations in the logarithmic scale formula.

- Was there an approval given by any Ethics committee or Institutional Review Board for this study? The authors only state that “All data used in this study were anonymous and therefore did not allow us to identify patients.” However, ethics or institutional approvals still need to be obtained for the use of the data, at least from the data owner, even if direct identifiable patient data was not used in the study.

- Line 203-205: Please specify during which period in the pandemic.

- Is there data regarding the predicted incidence vs. the actual incidence during the study time span?

- Since the study data was collected prior to the widescale deployment of COVID vaccination, the results can only be representative for an unvaccinated population. Since the age groups analyzed here are now included among those eligible to receive vaccination in most countries, this would be expected to significantly change the landscape for the upcoming start of the new education year. This should be discussed as a study limitation and further information should be provided regarding the vaccination program in Belgium, i.e., which age groups are now eligible, with particular emphasis on the student population, discussed in parallel with the vaccine uptake in this particular population group.

Reviewers' comments:

Reviewer's Responses to Questions

**Comments to the Author**

1. Is the manuscript technically sound, and do the data support the conclusions?

Reviewer #1: Partly

Reviewer #2: Yes

Reviewer #3: Yes

Reviewer #4: Yes

2. Has the statistical analysis been performed appropriately and rigorously? 

Reviewer #1: I Don't Know

Reviewer #2: Yes

Reviewer #3: Yes

Reviewer #4: Yes

3. Have the authors made all data underlying the findings in their manuscript fully available?

Reviewer #1: Yes

Reviewer #2: Yes

Reviewer #3: Yes

Reviewer #4: No

4. Is the manuscript presented in an intelligible fashion and written in standard English?

Reviewer #1: Yes

Reviewer #2: Yes

Reviewer #3: Yes

Reviewer #4: Yes

5. Review Comments to the Author

Reviewer #1: Reviewer comment

The authors efforts at demonstrating that, policies such as opening of higher education during an ongoing pandemic has consequences on the evolution of the pandemic is worth praising. However, the study is that of an ecological study design and the authors failed to convincingly show the association being described. Below are some of my concerns.

1. Generally, all the figures were difficult to understand due to poor readability of legends and quality of figures making understanding of the figures a problem.

2. Also, some figures lack the necessary description of axis eg; Fig 1, x-axis is missing

3. Difficult to understand the message from the map

4. Figure one was referenced in the introduction, second paragraph; I was not sure even after looking at figure one, if it is as a result of your analysis or copied from elsewhere. If copied, then the source of the fig, copyright issues need to be described. If it is as a result of your own analysis, then I would consider putting it in the results section and not introduction

5. Similarly, Figure two was placed in the introduction and discussed. I would have considered all discussion in the appropriate column.

Methods

6. Method section, paragraph 3, line 82 stated that “All data are obtained from Statbel, for the year 2016.” However, checking the open data section of statbel, it was obvious that it has upto date data to even 2021. The “Population by place of residence, nationality, marital status, age and sex” for 2020 is available. What informed the choice of 2016 dataset when 2020 dataset is available for direct comparison with 2020 COVID-19 data

7. Line 83 mentioned that, “The student ratio is presented in the upper right panel” of figure 4. However, this is not obvious from the figure 4

Results

8. Results section; line 134-139 stated that, “We observe that the age group of 18–29 years, followed by age group 30–39 years had the highest predicted incidence of COVID-19 cases, particularly from mid-September 2020 until the end of October 2020, in both Flanders and Wallo-Brux. From November 2020 onwards, the predicted incidence was highest within the group of those older than 80 years in both regions. This suggests that this epidemic wave started in the younger individuals, and that infection spread from the younger individuals to the elderly population.”

a. In as much as your assertion may be possible, there are other factors eg; relatively severe disease in the 80-year-old group from November onwards such that, the older age group are presenting for testing and the relatively asymptomatic/ mildly symptomatic younger age group are not presenting for testing.

b. Also we are you aware what variant of the virus was in circulation around November onwards, which may be different from that which was in circulation around September. The differences in the transmission and characteristics of the variants in circulation could also explain the observation

9. We need to understand how COVID-19 cases are tested in Belgium. Is it the situation as in many countries that symptomatic or people meeting a set of case definitions are tested.

10. The figures and graphs did not illustrate the association of increased incidence and the reopening of schools convincingly. It could also be due to the fact that the legends of the graphs are not legible, however, I was expecting a basic epi curve showing the time line of reopening of higher education and other possible interventions

Conclusion

11. Conclusion, lines 242-244 stated that “The results in this study provide insights into the association between reopening higher education in the academic year 2020-2021 and increased COVID-19 incidence in the Fall wave. However, this conclusion is not obvious from the graphs given.

12. Also, your study only addressed higher education. What about other schools which were non-higher education? Could the reopening of that also affect the findings?

13. The age group which was supposed to be in higher education as described 18-29, 30-39 also constitute the working age group. In most countries, reopening of higher education occurred with relaxing of restriction allowing the working group to engage in almost all possible economic activities. It is not convincing from your findings if the increase in incidence was mainly due to reopening of higher education.

14. Your study design is that of an ecological study, which comes with some limitations and biases.

a. Patient consultation habits, screening methods across the geographical regions may have an effect on the findings which need explanation

b. Diagnostic criteria over time and across geographical region could also play a part in the observed differences which needs explanation on how such might have been addressed or minimized.

Reviewer #2: Although the authors have done this study using COVID-19 data only for Belgium, this is an expedient work for the whole world during the ongoing pandemic situation. This study could help decision makers to make appropriate decision to lessen the burden of COVID-19. I would like to recommend this manuscript for possible publication after addressing the following minor issues:

1. The horizontal axis-label should be added in Fig 1.

2. The figure quality should be increased.

3. The legend of all figures should be clearly visualized.

Reviewer #3: article is well written with scientific explanation but its more focusing on the data associated with higher association but not gave explanation about opening of middle and lower school, inclusion or comparison with more data may give more accuracy to study

Reviewer #4: The manuscript is quite an interesting read and enjoyable! The analysis and results were presented well, as well as noted the limitations of the methods. The figures, however, are quite hard to read and needs to be provided in a higher resolution. I wasn't able to scrutinize the results because of that, so it will be good to have the figures re-sent.

6. PLOS authors have the option to publish the peer review history of their article (what does this mean?). If published, this will include your full peer review and any attached files.

Reviewer #1: No

Reviewer #2: No

Reviewer #3: No

Reviewer #4: No

---

## [Author Response · Author response to Decision Letter 0]

1 Oct 2021

First and foremost, we want to thank the reviewers for their critical assessment of our work and their constructive comments. We have addressed all of them and modi\fed the paper accordingly. Please find our detailed response in the file 'Response to Reviewers'.

---

## [Decision Letter · Decision Letter 1]

25 Nov 2021

PONE-D-21-17289R1The COVID-19 wave in Belgium during the Fall of 2020 and its association with higher educationPLOS ONE

Dear Dr. Natalia,

Thank you for submitting your manuscript to PLOS ONE. After careful consideration, we feel that it has merit but does not fully meet PLOS ONE’s publication criteria as it currently stands. Therefore, we invite you to submit a revised version of the manuscript that addresses the points raised during the review process. While most of the previous comments have been addressed during the revision process, some minor comments should further be addressed, as mentioned below:- In the newly added phrase on line 23, please correct the decimal indicator to a dot instead of a comma.

- In the newly added phrase: “First, incidences in the compulsory school population, monitored directly by the school system itself have been considerably lower than that of the general population, from the start of the school year (September 1, 2020) until February 2021.” – This might no longer be the case with surges of cases among school children in many European countries this autumn, now that delta has become the dominant variant. This could be briefly addressed as part of the discussion.

- In the newly added phrase “Some studies, conducted during the initial phase of the COVID-19 pandemic (March-April 2020), show that younger adults were more likely to leave their homes frequently [36] and take part in social activities such as parties, fitness, or casual hang-outs [37-40]” one further aspect could be considered, i.e., that younger children were generally “shielded” during the initial phase of the pandemic by their parents who continued to engage in work-related and social activities [see ref PMID 33489943]. It would be interesting to see how and if this translates into “shielding” of older children, i.e., university students engaged in higher education. This could be briefly addressed as part of the discussion.

- Please add a brief comment to the discussion section to address the change in transmission dynamics with the delta variant and to discuss how your results could be transposable or not to the new transmission scenario in areas where delta is now the main circulating variant.

We look forward to receiving your revised manuscript.

Kind regards,

Oana Săndulescu

Academic Editor

PLOS ONE

Journal Requirements:

Reviewers' comments:

Reviewer's Responses to Questions

**Comments to the Author**

1. If the authors have adequately addressed your comments raised in a previous round of review and you feel that this manuscript is now acceptable for publication, you may indicate that here to bypass the “Comments to the Author” section, enter your conflict of interest statement in the “Confidential to Editor” section, and submit your "Accept" recommendation.

Reviewer #1: All comments have been addressed

2. Is the manuscript technically sound, and do the data support the conclusions?

Reviewer #1: Yes

3. Has the statistical analysis been performed appropriately and rigorously? 

Reviewer #1: Yes

4. Have the authors made all data underlying the findings in their manuscript fully available?

Reviewer #1: No

5. Is the manuscript presented in an intelligible fashion and written in standard English?

Reviewer #1: Yes

6. Review Comments to the Author

Reviewer #1: The authors did a great job by incorporating all my comments and suggestions in to their revision of the manuscript

7. PLOS authors have the option to publish the peer review history of their article (what does this mean?). If published, this will include your full peer review and any attached files.

Reviewer #1: **Yes: **Benjamin D. Nuertey

---

## [Author Response · Author response to Decision Letter 1]

9 Jan 2022

Dear Reviewer(s)

Thank you for your comments on our revised submission. We have added more information of our analysis based on the current COVID-19 transmission. The details for each comment could be found in our 'Response to reviewer' file.

---

## [Editor Report · Decision Letter 2]

14 Feb 2022

The COVID-19 wave in Belgium during the Fall of 2020 and its association with higher education

PONE-D-21-17289R2

Dear Dr. Natalia,

We’re pleased to inform you that your manuscript has been judged scientifically suitable for publication and will be formally accepted for publication once it meets all outstanding technical requirements.

Kind regards,

Oana Săndulescu

Academic Editor

PLOS ONE

Additional Editor Comments (optional):

I thank the authors for carefully addressing all previous editor and reviewer comments.

---

## [Editor Report · Acceptance letter]

16 Feb 2022

PONE-D-21-17289R2 

The COVID-19 wave in Belgium during the Fall of 2020 and its association with higher education 

Dear Dr. Natalia:

I'm pleased to inform you that your manuscript has been deemed suitable for publication in PLOS ONE. Congratulations! Your manuscript is now with our production department. 

Kind regards, 

on behalf of

Dr. Oana Săndulescu 

Academic Editor

PLOS ONE